# A Simple Decentralized Cross-Entropy Method

**Zichen Zhang**[1,*]  **Jun Jin**[2]  **Martin Jagersand**[1]  **Jun Luo**[2,†]  **Dale Schuurmans**[1,†]

[1]University of Alberta   [2]Huawei Noah's Ark Lab   [†]equal advising
{zichen2,mj7,daes}@ualberta.ca   {jun.jin1,jun.luo1}@huawei.com

## Abstract

Cross-Entropy Method (CEM) is commonly used for planning in model-based reinforcement learning (MBRL) where a *centralized* approach is typically utilized to update the sampling distribution based on only the top-$k$ operations' results on samples. In this paper, we show that such a *centralized* approach makes CEM vulnerable to local optima, thus impairing its sample efficiency. To tackle this issue, we propose **Decent**ralized **CEM (DecentCEM)**, a simple but effective improvement over classical CEM, by using an ensemble of CEM instances running independently from one another, and each performing a local improvement of its own sampling distribution. We provide both theoretical and empirical analysis to demonstrate the effectiveness of this simple *decentralized* approach. We empirically show that, compared to the classical centralized approach using either a single or even a mixture of Gaussian distributions, our DecentCEM finds the global optimum much more consistently thus improves the sample efficiency. Furthermore, we plug in our DecentCEM in the planning problem of MBRL, and evaluate our approach in several continuous control environments, with comparison to the state-of-art CEM based MBRL approaches (PETS and POPLIN). Results show sample efficiency improvement by simply replacing the classical CEM module with our DecentCEM module, while only sacrificing a reasonable amount of computational cost. Lastly, we conduct ablation studies for more in-depth analysis. Code is available at https://github.com/vincentzhang/decentCEM.

## 1  Introduction

Model-based reinforcement learning (MBRL) uses a model as a proxy of the environment for planning actions in multiple steps. This paper studies planning in MBRL with a specific focus on the Cross-Entropy Method (CEM) [De Boer et al., 2005, Mannor et al., 2003], which is popular in MBRL due to its ease of use and strong empirical performance [Chua et al., 2018, Hafner et al., 2019, Wang and Ba, 2020, Zhang et al., 2021, Yang et al., 2020]. CEM is a stochastic, derivative-free optimization method. It uses a sampling distribution to generate imaginary trajectories of environment-agent interactions with the model. These trajectories are then ranked based on their returns computed from the rewards given by the model. The sampling distribution is updated to increase the likelihood of producing the top-$k$ trajectories with higher returns. These steps are iterated and eventually yield an improved distribution over the action sequences to guide the action execution in the real environment.

Despite the strong empirical performance of CEM for planning, it is prone to two problems: (1) lower sample efficiency as the dimensionality of solution space increases, and (2) the Gaussian distribution that is commonly used for sampling may cause the optimization to get stuck in local optima of multi-modal solution spaces commonly seen in real-world problems. Previous works addressing these problems either add gradient-based updates of the samples to optimize the parameters of CEM,

---

[*]Work partially done during Zichen's internship at Huawei Noah's Ark Lab.

36th Conference on Neural Information Processing Systems (NeurIPS 2022).

or adopt more expressive sampling distributions, such as using Gaussian Mixture Model [Okada and Taniguchi, 2020] or masked auto-regressive neural network [Hakhamaneshi et al., 2020]. Nevertheless, all CEM implementations to date are limited to a *centralized* formulation where the ranking step involves *all samples*. As analyzed below and in Section 3, such a centralized design makes CEM vulnerable to local optima and impairs its sample efficiency.

We propose **Decent**ralized **CEM (DecentCEM)**, a simple but effective improvement over classical CEM, to address the above problems. Rather than ranking *all samples*, as in the *centralized* design, our method distribute the sampling budget across an ensemble of CEM instances. These instances run independently from one another, and each performs a local improvement of its own sampling distribution based on the ranking of its generated samples. The best action is then aggregated by taking an $\arg\max$ among the solution of the instances. It recovers the conventional CEM when the number of instance is one.

We hypothesize that by shifting to this *decentralized* design, CEM can be less susceptible to premature convergence caused by the *centralized* ranking step. As illustrated in Fig. 1, the *centralized* sampling distribution exhibits a bias toward the sub-optimal solutions near top right, due to the *global* top-$k$ ranking. This bias would occur regardless of the family of distributions used. In comparison, a *decentralized* approach could maintain enough diversity thanks to its *local* top-$k$ ranking in each sampling instance.

Through a one-dimensional multi-modal optimization problem in Section 3, we show that DecentCEM empirically finds the global optimum more consistently than centralized CEM approaches that use either a single or a mixture of Gaussian distributions. Also we show that DecentCEM is theoretically sound that it converges almost surely to a local optimum. We further apply it to sequential decision making problems and use neural networks to parameterize the sampling distributions. Empirical results in several continuous control environments suggest that DecentCEM offers an effective mechanism to improve the sample efficiency over the baseline CEM under the same sample budget for planning.

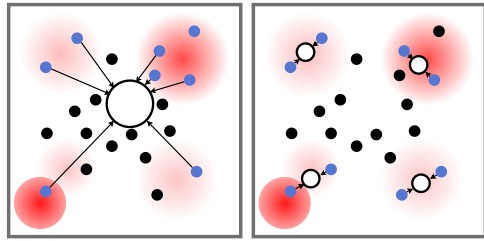

(a) Centralized CEM    (b) Decentralized CEM

Figure 1: Illustration of CEM approaches in optimization. Shades of red indicate relative value of the 2D optimization landscape: redder is better. and optimal solutions are near bottom left corner of the solution space. Blue dots ● are top-$k$ samples, and black dots ● are other samples. Open dots ○ represent the sampling distributions whose sizes indicate the number of generated samples.

## 2   Preliminaries

We consider a Markov Decision Process (MDP) specified by $(S, A, R, P, \gamma, d_0, T)$. $S \subset \mathbb{R}^{d_s}$ is the state space, $A \subset \mathbb{R}^{d_a}$ is the action space. $d_s, d_a$ are scalars denoting the dimensionality. $R : S \times A \to \mathbb{R}$ is the reward function that maps a state and action pair to a real-valued reward. $P(s'|s, a) : S \times A \times S \to \mathbb{R}^+$ is the transition probability from a state and action pair $s, a$ to the next state $s'$. $\gamma \in [0, 1]$ is the discount factor. $d_0$ denotes the distribution of the initial state $s_0$. At time step $t$, the agent receives a state $s_t$ [2] and takes an action $a_t$ according to a policy $\pi(\cdot|s)$ that maps the state to a probability distribution over the action space. The environment transitions to the next state $s_{t+1} \sim P(\cdot|s_t, a_t)$ and gives a reward $r_t = R(s_t, a_t)$ to the agent. Following the settings from CEM-based MBRL papers (Sec. 6.1), we assume that the reward function is deterministic (a mild assumption [Agarwal et al., 2019]) and known. Note that they are not fundamental limitations of CEM and are adopted here so as to be consistent with the literature. The return $G_t = \sum_{i=0}^{T} \gamma^i r_{t+i}$, is the sum of discounted reward within an episode length of $T$. The agent aims to find a policy $\pi$ that maximizes the expected return. We denote the learned model in MBRL as $f_\omega(\cdot|s, a)$, which is parameterized by $\omega$ and approximates $P(\cdot|s, a)$.

**Planning with the Cross Entropy Method**   Planning in MBRL is about leveraging the model to find the best action in terms of its return. Model-Predictive-Control (MPC) performs decision-time

---

[2]We assume full observability, i.e. agent has access to the state.

planning at each time step up to a horizon to find the optimal action sequence:

$$\pi_{\text{MPC}}(s_t) = \underset{a_{t:t+H-1}}{\arg\max} \mathbb{E}[\Sigma_{i=0}^{H-1} \gamma^i r(s_{t+i}, a_{t+i}) + \gamma^H V(s_H)] \tag{1}$$

where $H$ is the planning horizon, $a_{t:t+H-1}$ denotes the action sequence from time step $t$ to $t+H-1$, and $V(s_H)$ is the terminal value function at the end of the horizon. The first action in this sequence is executed and the rest are discarded. The agent then re-plans at the next time step.

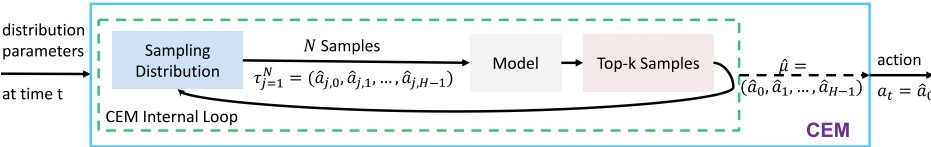

Figure 2: Cross Entropy Method (CEM) for Planning in MBRL

The Cross-Entropy Method (CEM) is a gradient-free optimization method that can be used for solving Eq. (1). The workflow is shown in Fig. 2. CEM planning starts by generating $N$ samples $\{\tau_j\}_{j=1}^{N} = \{(\hat{a}_{j,0}, \hat{a}_{j,1}, \cdots, \hat{a}_{j,H-1})\}_{j=1}^{N}$ from an initial sampling distribution $g_\phi(\tau)$ parameterized by $\phi$, where each sample $\tau_j$ is an action sequence from the current time step up to the planning horizon $H$. The domain of $g_\phi(\tau)$ has a dimension of $d_\tau = d_a H$.

Using a model $f$, CEM generates imaginary rollouts based on the action sequence $\{\tau_j\}$ (in the case of a stochastic model) and estimate the associated value $v(\tau_j) = \mathbb{E}[\Sigma_{i=0}^{H-1} \gamma^i r(s_{j,i}, a_{j,i})]$ where $s_{j,0}$ is the current state $s$ and $s_{j,i+1} \sim f(\cdot|s_{j,i}, a_{j,i})$. The terminal value $\gamma^H V(s_{j,H})$ is omitted here following the convention in the CEM planning literature but the MPC performance can be further improved if paired with an accurate value predictor [Bertsekas, 2005, Lowrey et al., 2019]. The sampling distribution is then updated by fitting to the current top-$k$ samples in terms of their value estimates $v(\tau_j)$, using the Maximum Likelihood Estimation (MLE) which solves:

$$\phi' = \underset{\phi}{\arg\max} \sum_{j=1}^{N} \mathbb{1}(v(\tau_j) \geq v_{\text{th}}) \log g_\phi(\tau_j) \tag{2}$$

where $v_{\text{th}}$ is the threshold equal to the value of the $k$-th best sample and $\mathbb{1}(\cdot)$ is the indicator function. In practice, the update to the distribution parameters are smoothed by $\phi^{l+1} = \alpha \phi' + (1-\alpha)\phi^l$ where $\alpha \in [0,1]$ is a smoothing parameter that balances between the solution to Eq. (2) and the parameter at the current internal iteration $l$. CEM repeats this process of sampling and distribution update in an inner-loop, until it reaches the stopping condition. In practice, it is stopped when either a maximum number of iterations has been reached or the parameters have not changed for a few iterations. The output of CEM is an action sequence, typically set as the expectation[3] of the most recent sampling distribution for uni-modal distributions such as Gaussians $\hat{\mu} = \mathbb{E}(g_\phi) = (\hat{a}_0, \hat{a}_1, \cdots, \hat{a}_{H-1})$.

**Choices of Sampling Distributions in CEM:** A common choice is a multivariate Gaussian distribution under which Eq.(2) has an analytical solution. But the uni-modal nature of Gaussian makes it inadequate in solving multi-modal optimization that often occurs in MBRL. To increase the capacity of the distribution, a Gaussian Mixture Model (GMM) can be used [Okada and Taniguchi, 2020]. We denote such an approach as *CEM-GMM*. Going forward, we use *CEM* to refer to the vanilla version that employs a Gaussian distribution. Computationally, the major difference between *CEM* and *CEM-GMM* is that the distribution update in *CEM-GMM* is more computation-intensive since it solves for more parameters. Detailed steps can be found in Okada and Taniguchi [2020].

## 3  Decentralized CEM

In this section, we first introduce the formulation of the proposed *decentralized* approach called the **Decent**ralized **CEM (DecentCEM)**. Then we illustrate the intuition behind the proposed approach using a one-dimensional synthetic multi-modal optimization example where we show the issues of the existing CEM methods and how they can be addressed by DecentCEM.

---

[3]Other options are discussed in Appendix A.2

**Formulation of DecentCEM**  DecentCEM is composed of an ensemble of $M$ CEM instances indexed by $i$, each having its own sampling distributions $g_{\phi_i}$. They can be described by a set of distribution parameters $\Phi = \{\phi_i\}_{i=1}^M$. Each instance $i$ manages its own sampling and distribution update by the steps described in Section 2, independently from other instances.

Note that the $N$ samples and $k$ elites are evenly split among the $M$ instances. The top-$\frac{k}{M}$ sample sets are decentralized and managed by each instance independently whereas the centralized approach only keeps one set of top-$k$ samples regardless of the distribution family used.

After the stopping condition is reached for all instances, the final sampling distribution is taken as the best distribution in the set $\Phi$ according to (the $\arg\max$ uses a deterministic tie-breaking):

$$\phi_{\text{DecentCEM}} = \underset{\phi_i \in \Phi}{\arg\max}\, \mathbb{E}_{\phi_i}[v(x)] \approx \underset{\phi_i \in \Phi}{\arg\max} \sum_{j=1}^{\frac{N}{M}} v(\tau_{i,j}) \tag{3}$$

where $\mathbb{E}_{\phi_i}[v(x)]$ denotes the expectation with respect to the distribution $g_{\phi_i}$, approximated by the sample mean of $\frac{N}{M}$ samples. When $M = 1$, it recovers the conventional CEM.

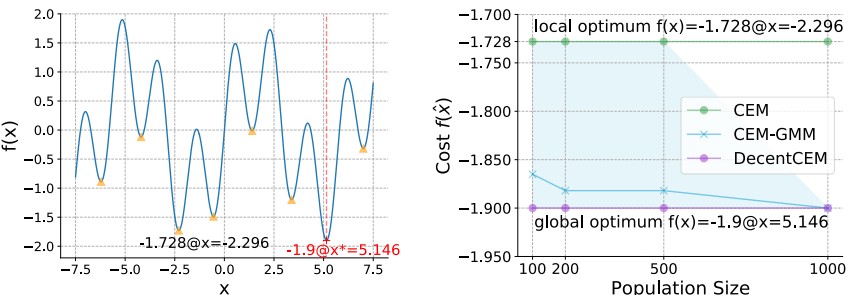

Figure 3: Left: The objective function in a 1D optimization task. Right: Comparison of the proposed *DecentCEM* method to *CEM* and *CEM-GMM*, wherein the line and the shaded region denote the mean and the min/max cost from 10 independent runs. $\hat{x}$: solution of each method.

**Motivational Example**  Consider a one dimensional multi-modal optimization problem shown in Fig. 3 (Left): $\arg\min_x \sin(x) + \sin(10x/3), -7.5 \le x \le 7.5$. There are eight local optima, including one global optimum $f(x^*) = -1.9$ where $x^* = 5.146$. This objective function mimics the RL value landscape that has many local optima, as shown by Wang and Ba [2020]. This optimization problem is "easy" in the sense that a grid search over the domain can get us a solution close to the global optimum. However, only our proposed *DecentCEM* method successfully converges to the global optimum consistently under varying population size (i.e., number of samples) and random runs, as shown in Fig. 3 (Right). For a fair comparison, hyperparameter search has been conducted on all methods for each population size (Appendix A).

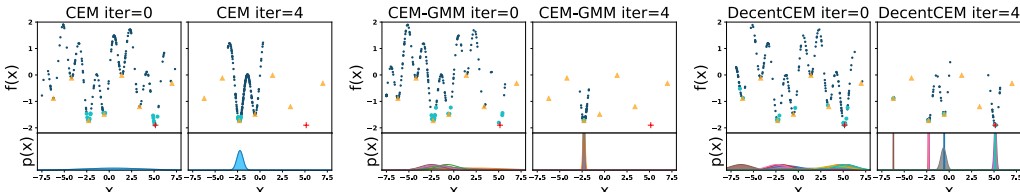

Figure 4: How the sampling distributions evolve in the 1D optimization task, *after* the specified iteration. Symbols include samples ●, elites ●, local optima ▲, global optimum +. 2nd row in each figure shows the weighted p.d.f of individual distribution. Population size: 200.

Both *CEM-GMM* and the proposed *DecentCEM* are equipped with multiple sampling distributions. The fact that *CEM-GMM* is outperformed by *DecentCEM* may appear surprising. To gain some insights, we illustrate in Fig. 4 how the sampling distribution evolves during the iterative update (more details in Fig. 10 in Appendix). *CEM* updated the unimodal distribution toward a local optimum despite seeing the global optimum. *CEM-GMM* appears to have a similar issue. During MLE on the top-$k$ samples, it moved most distribution components towards the same local optimum which quickly

led to mode collapse. On the contrary, *DecentCEM* successfully escaped the local optima thanks to its independent distribution update over *decentralized* top-$k$ samples and was able to maintain a decent diversity among the distributions.

GMM suits density estimation problems like distribution-based clustering where the samples are drawn from a *fixed true* distribution that can be represented by multi-modal Gaussians. However, in CEM for optimization, exploration is coupled with density estimation: the sampling distribution in CEM is *not fixed* but rather gets updated iteratively toward the top-$k$ samples. And the "true" distribution in optimization puts uniform non-zero densities to the global optima and zero densities everywhere else. When there is a unique global optimum, it degenerates into a Dirac measure that assigns the entire density to the optimum. Density estimation of such a distribution only needs one Gaussian but the exploration is challenging. In other words, the conditions for GMM to work well are not necessarily met when used as the sampling distribution in *CEM*. *CEM-GMM* is subject to mode collapse during the iterative top-$k$ greedification, causing premature convergence, as observed in Fig 4. In comparison, our proposed decentralized approach takes care of the exploration aspect by running multiple CEM instances independently, each performing its own local improvement. This is shown to be effective from this optimization example and the benchmark results in Section 6. *CEM-GMM* only consistently converge to the global optimum when we increase the population size to the maximum 1,000 which causes expensive computations. Our proposed *DecentCEM* runs more than 100 times faster than *CEM-GMM* at this population size, shown in Table A.3 in Appendix.

**Convergence of DecentCEM**    The convergence of DecentCEM requires the following assumption:

**Assumption 1.** *Let $M$ be the number of instances in DecentCEM and each instance has a sample size of $\frac{N_k}{M}$ where $N_k$ is the total number of samples that satisfies $N_k = \Theta(k^\beta)$ where $\beta > \max\{0, 1-2\lambda\}$. $M$ and $\lambda$ are some positive constant and $0 < M < N_k \ \forall \ k$.*

Here the assumption of the total sample size follows the same one as in the CEM literature. Other such standard assumptions are summarized in Appendix H.

**Theorem 3.1** (Convergence of DecentCEM). *If a CEM instance described in Algorithm 3 converges, and we decentralize it by evenly dividing its sample size $N_k$ into $M$ CEM instances which satisfies the Assumption 1, then the resulting DecentCEM converges almost surely to the best solution of the individual instances.*

*Proof.* (sketch) We show that the previous convergence result of CEM [Hu et al., 2011] extends to DecentCEM under the same sample budget. The key observation is that the convergence property of each CEM instance still holds since the number of samples in each instance is only changed by a constant factor, i.e., number of instances. Each CEM instance converges to a local optimum. The convergence of DecentCEM to the best solution comes from the $\arg\max$ operator and applying the strong law of large numbers. The detailed proof is left to Appendix H. □

## 4    DecentCEM for Planning

In this section, we develop two instantiations of DecentCEM for planning in MBRL where the sampling distributions are parameterized by policy networks. For the dynamics model learning, we adopt the ensemble of probabilistic neural networks proposed in [Chua et al., 2018]. Each network predicts the mean and diagonal covariance of a Gaussian distribution and is trained by minimizing the negative log likelihood. Our work focuses on the planning aspect and refer to Chua et al. [2018] for further details on the model.

**CEM Planning with a Policy Network**    In MBRL, CEM is applied to every state separately to solve the optimization problem stated in Eq. (1). The sampling distribution is typically initialized to a fixed distribution at the beginning of every episode [Okada and Taniguchi, 2020, Pinneri et al., 2020], or more frequently at every time step [Hafner et al., 2019]. Such initialization schemes are sample inefficient since there is no mechanism that allows the information of the high-value region in the value space of one state to generalize to nearby states. Also, the information is discarded after the initialization. It is hence difficult to scale the approach to higher dimensional solution spaces, present in many continuous control environments. Wang and Ba [2020] proposed to use a policy network in CEM planning that helped to mitigate the issues above.

They developed two methods: *POPLIN-A* that plans in the action space, and *POPLIN-P* that plans in the parameter space of the policy network. In *POPLIN-A*, the policy network is used to learn to output the mean of a Gaussian sampling distribution of actions. In *POPLIN-P*, the policy network parameters serve as the initialization of the mean of the sampling distribution of parameters. The improved policy network can then be used to generate an action. They show that when compared to the vanilla method of using a fixed sampling distribution in the action space, both modes of CEM planning with such a learned distribution perform better. The same principle of combining a policy network with CEM can be applied to the DecentCEM approach as well, which we describe next.

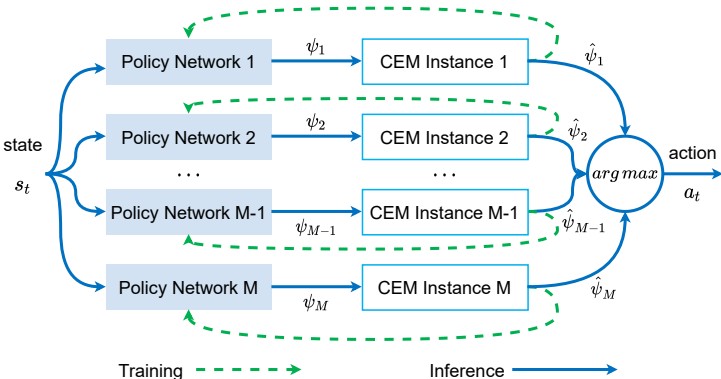

Figure 5: The architecture of DecentCEM planning with $M$ CEM instances. $\psi_i = \phi_i$ for planning in action space and $\psi_i = \theta_i$ for planning in policy network parameter space with the instance index $i \in \{1, \cdots, M\}$.

**DecentCEM Planning with an Ensemble of Policy Networks** For better sample efficiency in MBRL setting, we extend DecentCEM to use an ensemble of policy networks to learn the sampling distributions in the CEM instances. Similar to the *POPLIN* variations, we develop two instantiations of *DecentCEM*, namely *DecentCEM-A* and *DecentCEM-P*. The architecture of the proposed algorithm is illustrated in Fig. 5. *DecentCEM-A* plans in the action space. It consists of an ensemble of policy networks followed by CEM instances. Each policy network takes the current state $s_t$ as input, outputs the parameters $\phi_i$ of the sampling distribution for CEM instance $i$ in the action space. There is no fundamental difference from the DecentCEM formulation in Section 3 except that the initialization of sampling distributions is learned by the policy networks rather than a fixed distribution.

The second instantiation *DecentCEM-P* plans in the parameter space of the policy network. The output of each policy network is the network parameter $\theta_i$. The initial sampling distribution of CEM instance $i$ is a Gaussian distribution over the policy parameter space with the mean at $\theta_i$. In the $\arg\max$ operation in Eq. (3), the sample $\tau_{i,j}$ denotes the $j$-th parameter sample from the distribution for CEM instance $i$. Its value is approximated by the model-predicted value of the action sequence generated from the policy network with the parameters $\tau_{i,j}$.

The ensemble of policy networks in both instantiations *DecentCEM-A* and *DecentCEM-P* are initialized with random weights, which is empirically found to be adequate to ensure that the output of the networks do not collapse into the same distribution (Sec. 6.2 and Appendix F).

**Training the Policy Networks in DecentCEM** When planning in action space, the policy networks are trained by behavior cloning, similar to the scheme in *POPLIN* [Wang and Ba, 2020]. Denote the first action in the planned action sequence at time step $t$ by the $i$-th CEM instance as $\hat{a}_{t,i}$, the $i$-th policy network is trained to mimic $\hat{a}_{t,i}$ and the training objective is $\min_{\theta_i} \mathbb{E}_{s_t, \hat{a}_{t,i} \sim D_i} \|a_{\theta_i}(s_t) - \hat{a}_{t,i}\|^2$ where $D_i$ denotes the replay buffer with the state and action pairs $(s_t, \hat{a}_{t,i})$. $a_{\theta_i}(s_t)$ is the action prediction at state $s_t$ from the policy network parameterized by $\theta_i$.

While the above training scheme can be applied to both planning in action space and parameter space, we follow the *setting parameter average* (AVG) [Wang and Ba, 2020] training scheme when planning in parameter space. The parameter is updated as $\theta_i = \theta_i + \frac{1}{|D_i|} \sum_{\delta_i \in D_i} \delta_i$ where $D_i = \{\delta_i\}$ is a dataset of policy network parameter updates planned from the $i$-th CEM instance previously. It is more effective than behavior cloning based on the experimental result reported by Wang and Ba [2020] and our own preliminary experiments.

Note that each policy network in the ensemble is trained independently from the data observed by its corresponding CEM instance rather than from the aggregated result after taking the $\arg\max$. This allows for enough diversity among the instances. More importantly, it increases the size of the training dataset for the policy networks compared to the approach taken in *POPLIN*. For example, with an ensemble of $M$ instances, there would be $M$ training data samples available from one real environment interaction, compared to the one data sample in *POPLIN-A/P*. As a result, *DecentCEM* is able to train larger policy networks than is otherwise possible, shown in Sec. 6.2 and Appendix F.

## 5  Related Work

We limit the scope of related works to CEM planning methods, which is one of the broad class of planning algorithms in MBRL. For a review of different families of planning algorithms, the readers are referred to Wang et al. [2019]. Vanilla CEM planning in action space with a single Gaussian distribution has been adopted as the planning method for both simulated and real-world robot control [Chua et al., 2018, Finn and Levine, 2017, Ebert et al., 2018, Hafner et al., 2019, Yang et al., 2020, Zhang et al., 2021]. Previous attempts to improving the performance of CEM-based planning can be grouped into two types of approaches. **The first type** includes CEM in a hybrid of CEM+X where "X" is some other component or algorithm. POPLIN [Wang and Ba, 2020] is a prominent example where "X" is a policy network that learns a state conditioned distribution that initializes the subsequent CEM process. Another common choice of "X" is gradient-based adjustment of the samples drawn in CEM. GradCEM [Bharadhwaj et al., 2020] adjusts the samples in each iteration of CEM by taking gradient ascent of the return estimate w.r.t the actions. CEM-RL [Pourchot and Sigaud, 2019] also combines CEM with gradient based updates from RL algorithms but the samples are in the parameter space of the actor network. To improve computational efficiency, Lee et al. [2020] proposes an asynchronous version of CEM-RL where each CEM instance updates the sampling distribution asynchronously.

**The second type** of approach aims at improving CEM itself. Amos and Yarats [2020] proposes a fully-differentiable version of CEM called *DCEM*. The key is to make the top-$k$ selection in CEM differentiable such that the entire CEM module can be trained in an end-to-end fashion. Despite cutting down the number of samples needed in CEM, this method does not beat the vanilla CEM in benchmark test. *GACEM* [Hakhamaneshi et al., 2020] increase the capacity of the sampling distribution by replacing the Gaussian distribution with an auto-regressive neural network. This change allows CEM to perform search in multi-modal solution space but it is only verified in toy examples and its computation seems too high to be scaled to MBRL tasks. Another method that increases the capacity of the sampling distribution is *PaETS* [Okada and Taniguchi, 2020] that uses a GMM with CEM. It is the approach that we followed for our *CEM-GMM* implementation. The running time results in the optimization task in Sec.3 shows that it is computationally heavier than the *CEM* and *DecentCEM* methods, limiting its use in complex environments. Macua et al. [2015] proposed a "distributed" CEM that is similar in spirit to our method in that they used multiple sampling distributions and applied the top-$k$ selection locally to samples from each instance. However, their instances are cooperative as opposed to being independent as in our work. They applied "collaborative smoothed projection steps" to update each sampling distribution as an average of its neighboring instances including itself. The updating procedure is more complicated than our proposed method and proper network topology of the instances is needed: a naive approach of updating according to all instances will lead to mode collapse since the resulting sampling distributions will be identical. The method was tested in toy optimization examples only. Overall, this second type of approach did not outperform vanilla CEM, a phenomenon that motivated our move to a decentralized formulation.

## 6  Experiments

We evaluate the proposed *DecentCEM* methods in simulated environments with continuous action space. The experimental evaluation is mainly set up to understand if *DecentCEM* improves the performance and sample efficiency over conventional CEM approaches.

### 6.1  Benchmark Setup

***Environments*** We run the benchmark in a set of OpenAI Gym [Brockman et al., 2016] and Mu-JoCo [Todorov et al., 2012] environments commonly used in the MBRL literature: Pendulum,

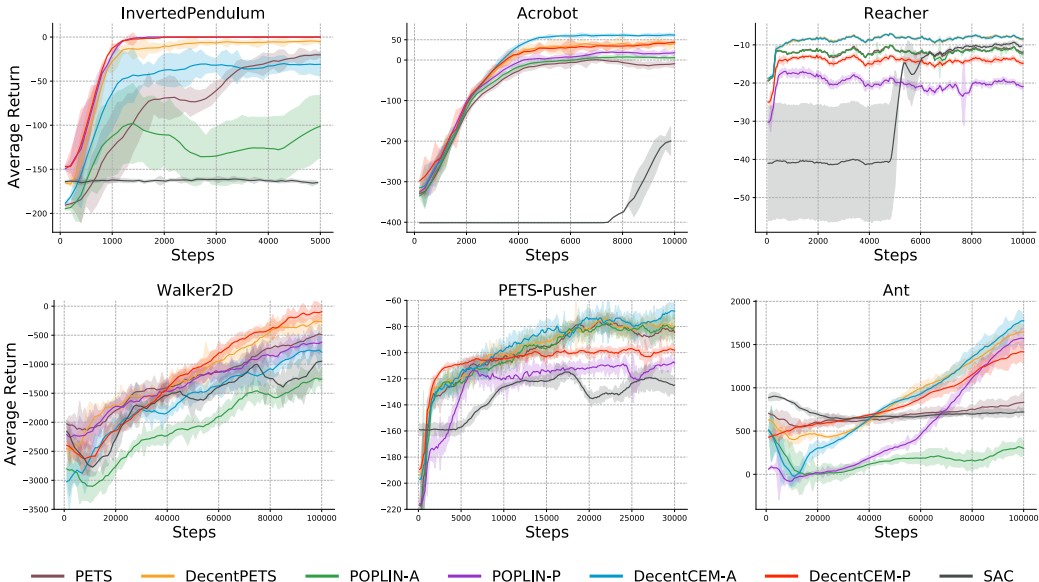

Figure 6: Learning curves of the proposed *DecentCEM* methods and the baselines on continuous control environments. The line and the shaded region shows the mean and standard error of evaluation results from 5 training runs with different random seeds.

InvertedPendulum, Cartpole, Acrobot, FixedSwimmer[4], Reacher, Hopper, Walker2D, HalfCheetah, PETS-Reacher3D, PETS-HalfCheetah, PETS-Pusher, Ant. The three environments prefixed by "PETS" are proposed by Chua et al. [2018]. Note that MBRL algorithms often make different assumptions about the dynamics model or the reward function. Their benchmark environments are often modified from the original OpenAI gym environments such that the respective algorithm is runnable. Whenever possible, we inherit the same environment setup from that of the respective baseline methods. This is so that the comparison against the baselines is fair. More details on the environments and their reward functions are in Appendix B.

***Algorithms*** The baseline algorithms are *PETS* [Chua et al., 2018] and *POPLIN* [Wang and Ba, 2020]. *PETS* uses CEM with a single Gaussian distribution for planning. The *POPLIN* algorithm combines a single policy network with CEM. As described in Sec. 4, *POPLIN* comes with two modes: *POPLIN-A* and *POPLIN-P* with the suffix "A" denotes planning in action space and "P" for the network parameter space. We reuse default hyperparameters for these algorithms from the original papers if not mentioned specifically. For our proposed methods, we include two variations *DecentCEM-A* and *DecentCEM-P* as described in Sec. 4 where the suffix carries the same meaning as in *POPLIN-A/P*. The ensemble size of *DecentCEM-A/P* as well as detailed hyperparameters for all algorithms are listed in the Appendix D.2. We also included Decentralized *PETS*, denoted by *DecentPETS*. All MBRL algorithms studied in this benchmark use the same ensemble networks proposed by Chua et al. [2018] for model learning. And a model-free RL baseline SAC [Haarnoja et al., 2018] was included.

***Evaluation Protocol*** The learning curve shows the mean and standard error of the test performance out of 5 independent training runs. The test performance is an average return of 5 episodes of the *evaluation* environment, evaluated at every training episode. At the beginning of each training run, the evaluation environment is initialized with a fixed random seed such that the *evaluation* environments are consistent across different methods and multiple runs to make it a fair comparison. All experiments were conducted using Tesla V100 Volta GPUs.

## 6.2 Results

**Learning Curves** The learning curves of the algorithms are shown in Fig. 6 for InvertedPendulum, Acrobot, Reacher, Walker2D, PETS-Pusher and Ant, sorted by the state and action dimensionality

---

[4]a modified version of the original Gym Swimmer environment where the velocity sensor on the neck is moved to the head. This fix was proposed by Wang and Ba [2020]

of task. The curves are smoothed with a sliding window of 10 data points. The full results for all environments are included in Appendix E. *We can observe two main patterns from the results.* One pattern was that in most environments, the *decentralized* methods *DecentPETS, DecentCEM-A/P* either matched or outperformed their counterpart that took a *centralized* approach. In fact, the former can be seen as a generalization of the later, by an additional hyperparameter that controls the ensemble size with size one recovering the centralized approach. The optimal ensemble size depends on the task. This additional hyperparameter offers flexibility in fine-tuning CEM for individual domains. For instance in Fig. 6, for the P-mode where the planning is performed in policy parameter space, an ensemble size of larger than one works better in most environments while an ensemble size of one works better in Ant. The other pattern was that using policy networks to learn the sampling distribution in general helped improving the performance of *centralized* CEM but not necessarily in *decentralized* formulation. This is perhaps due to that the added exploration from multiple instances makes it possible to identify good solutions in some environments. Using a policy net in such case may hinder the exploration due to overfitting to previous actions.

**Ablation Study** A natural question to ask about the *DecentCEM-A/P* methods is whether the increased performance is from the larger number of neural network parameters. We added two variations of the *POPLIN* baselines where a bigger policy network was used. The number of the network parameters was equivalent to that of the ensemble of policy networks in *DecentCEM-A/P*. We show the comparison in Reacher(2) and PETS-Pusher(7) (action dimension in parenthesis) in Fig. 7. In both action space and parameter space planning, a bigger policy network in *POPLIN* either did not help or significantly impaired the performance (see the *POPLIN-P* results in reacher and PETS-Pusher). This is expected since unlike *DecentCEM*, the training data in *POPLIN* do not scale with the size of the policy network, as explained at the end of Sec. 4.

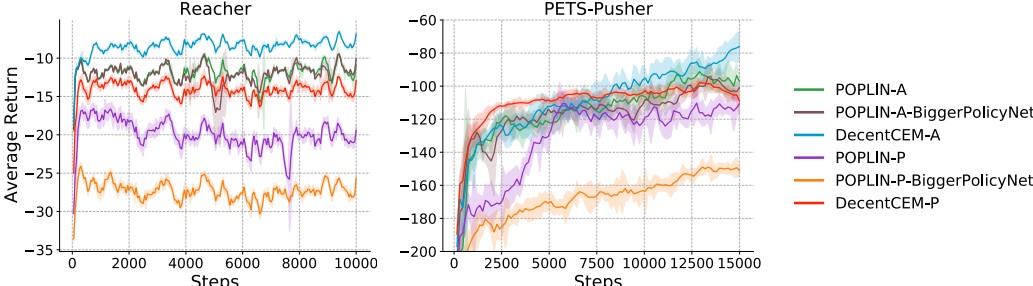

Figure 7: Ablation study on the policy network size where *POPLIN-A&P* have a bigger policy network equivalent in the total number of network weights to their *DecentCEM* counterparts.

Next, we look into the impact of ensemble size. Fig. 8 shows the learning curves of different ensemble sizes in the Reacher environments for action space planning (left), parameter space planning (middle) and planning without policy networks (right). Since we fix the total number of samples the same across the methods, the larger the ensemble size is, the fewer samples that each instance has access to. As the ensemble size goes up, we are trading off the accuracy of the sample mean for better exploration by more instances. Varying this hyperparameter allows us to find the best trade-off, as can be observed from the figure where increasing the ensemble size beyond a sweet spot yields diminishing or worse returns.

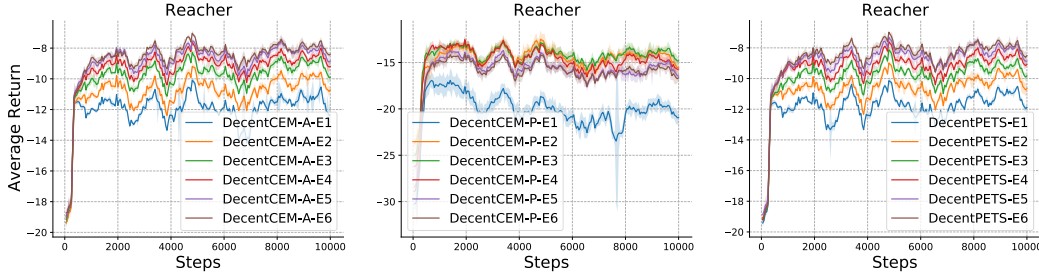

Figure 8: Ablation study on the ensemble size in *DecentCEM-A/P* and *DecentPETS*, e.g. E2 denotes an ensemble with 2 instances. The total number of samples during planning are the same across all variations.

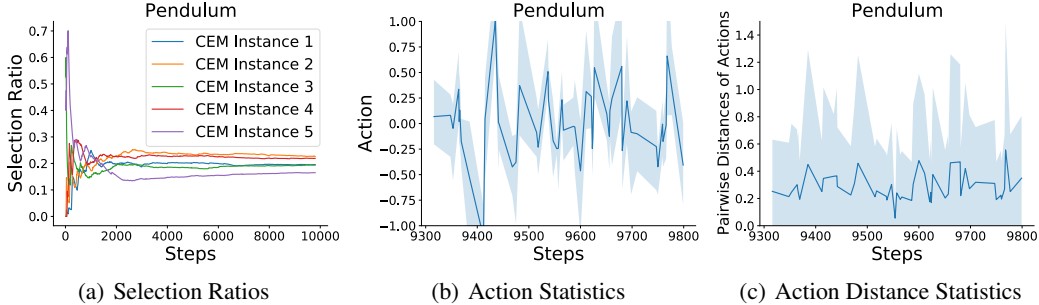

|                      |                      |                      |
|:--------------------:|:--------------------:|:--------------------:|
| (a) Selection Ratios | (b) Action Statistics | (c) Action Distance Statistics |

Figure 9: Ablation of ensemble diversity in Pendulum during training of *DecentCEM-A* with 5 instances. (a) Cumulative selection ratio of each CEM instance. (b)(c) Statistics of the actions and pairwise action distances of the instances, respectively. The line and shaded region represent the mean and min/max.

Figure 9(a) shows the cumulative selection ratio of each CEM instance during training of *DecentCEM-A* with an ensemble size of 5. It suggests that the random initialization of the policy network is sufficient to avoid mode collapse. We also plot the actions and pairwise action distances of the instances in Figure 9(b) and 9(c). For visual clarity, we show a time segment toward the end of the training rather than all the 10k steps. *DecentCEM-A* has maintained enough diversity in the instances even toward the end of the training. *DecentCEM-P* and *DecentPETS* share similar trends. The plots of their results along with other ablation results are included in Appendix F.

# 7    Conclusion and Future Work

In this paper, we study CEM planning in the context of continuous-action MBRL. We propose a novel *decentralized* formulation of CEM named *DecentCEM*, which generalizes CEM to run multiple independent instances and recovers the conventional CEM when the number of instances is one. We illustrate the strengths of the proposed DecentCEM approach in a motivational one-dimensional optimization task and show how it fundamentally differs from the CEM approach that uses a Gaussian or GMM. We also show that DecentCEM has almost sure convergence to a local optimum. We extend the proposed approach to MBRL by plugging in the *decentralized* CEM into three previous CEM-based methods: *PETS*, *POPLIN-A*, *POPLIN-P*. We show their efficacy in benchmark control tasks and ablations studies.

There is a gap between the convergence result and practice where the theory assumes that the number of samples grow polynomially with the iterations whereas a constant sample size is commonly used in practice including our work. Investigating the convergence properties of *CEM* under a constant sample size makes an interesting direction for future work. Another interesting direction to pursue is finite-time analysis of CEM under both *centralized* and *decentralized* formulations. In addition, the implementation has room for optimization: the instances currently run serially but can be improved by a parallel implementation to take advantage of the parallelism of the ensemble.

## Acknowledgments and Disclosure of Funding

We would like to thank the anonymous reviewers for their time and valuable suggestions. Zichen Zhang would like to thank Jincheng Mei for helpful discussions in the convergence analysis and Richard Sutton for raising a question about the model learning. This work was partially done during Zichen Zhang's internship at Huawei Noah's Ark Lab. Zichen Zhang gratefully acknowledges the financial support by an NSERC CGSD scholarship and an Alberta Innovates PhD scholarship. He is thankful for the compute resources generously provided by Digital Research Alliance of Canada (and formerly Compute Canada), which is sponsored through the accounts of Martin Jagersand and Dale Schuurmans. Dale Schuurmans gratefully acknowledges funding from the Canada CIFAR AI Chairs Program, Amii and NSERC.

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
