# Appendix

## A  Details of the Motivational Example

### A.1  Setup and Running Time

For a fair comparison of the three methods *CEM*, *CEM-GMM* and *DecentCEM*, we performed a hyperparameter search for all. The list of hyperparmeters are summarized in Table A.1 and the best performing hyperparameters for each method under each population size are shown in Table A.2. These hyperparameters were what the data in Fig. 3 were based on. Note that the top percentage of samples "Elite Ratio" (in Table A.1) was used in the implementation instead of top-$k$ but they are equivalent. The running time are included in Table A.3.

Table A.1: Hyperparameters

| Algorithm | Parameter | Value |
|---|---|---|
| Shared Parameters | Total Sample Size
Elite Ratio
$\alpha$: Smoothing Ratio
$\epsilon$: Minimum Variance Threshold
Maximum Number of Iterations | 100, 200, 500, 1000
0.1
0.1
1e-3
100 |
| CEM-GMM | $M$: Number of Mixture Components
$\kappa$: Weights of Entropy Regularizer
$r$: Return Mode | 3,5,8,10
0.25, 0.5
's': mean of the mixture component
 sampled based on their weights
'm': mean of the component that
 achieves the minimum cost |
| DecentCEM | $E$: Number of Instances in the Ensemble | 3,5,8,10 |

Table A.2: Best Hyper-Parameter

| | Total Sample Size | | | |
|---|---|---|---|---|
| | 100 | 200 | 500 | 1000 |
| CEM-GMM | $M = 10$
$\kappa = 0.25,$
$r =$ 'm' | $M = 8$
$\kappa = 0.5$
$r =$ 'm' | $M = 8$
$\kappa = 0.25$
$r =$ 'm' | $M = 8$
$\kappa = 0.5$
$r =$ 's' |
| DecentCEM | $E = 10$ | $E = 10$ | $E = 10$ | $E = 8$ |

Table A.3: Total Time of 10 Runs (in seconds)

| | Total Sample Size | | | |
|---|---|---|---|---|
| | 100 | 200 | 500 | 1000 |
| CEM | 0.079 | 0.093 | 0.165 | 0.318 |
| CEM-GMM | 7.322 | 11.500 | 24.431 | 59.844 |
| DecentCEM | 0.407 | 0.420 | 0.506 | 0.545 |

### A.2  Output of CEM Approaches

In terms of the output of CEM approaches, there exist different options in the literature. The most common option is to return the sample in the domain that corresponds to the highest probability density in the final sampling distribution. It is the mean in the case of Gaussian and the mode with the highest probability density in the case of GMM. One can also draw a sample from the final sampling distribution [Okada and Taniguchi, 2020] and return it. Another option is to return the best sample observed [Pinneri et al., 2020]. The best option among the three may be application dependent. It has been observed that in many applications, the sequence of sampling distributions numerically converges to a deterministic one [De Boer et al., 2005], in which case the first two options are identical.

Table A.4: The setup of the environments. The number in the bracket in the "Environment" column denotes the source of this environment: [1] refers to the benchmark paper from Wang et al. [2019]; [2] denotes PETS [Chua et al., 2018]. In the reward functions, $\mathbf{d}_t$ denotes the vector between the end effector to the target position. $z_t$ denotes the height of the robot. $\|\mathbf{v}\|_1$ and $\|\mathbf{v}\|_2$ denote the 1-norm and 2-norm of vector $\mathbf{v}$, respectively. In PETS-Pusher, $\mathbf{d}_{1,t}$ is the vector between the object position and the goal and $\mathbf{d}_{2,t}$ denotes the vector between the object position and the end effector.

| Environment | $|S|$ | $|A|$ | Episode Length | Reward Function |
|---|---|---|---|---|
| Pendulum | 3 | 1 | 200 | $\theta_t^2 + 0.1\dot{\theta}_t^2 + 0.001a_t^2$ |
| InvertedPendulum [1] | 4 | 1 | 100 | $-\theta_t^2$ |
| Cartpole [1] | 4 | 1 | 200 | $\cos\theta_t - 0.01x_t^2$ |
| Acrobot [1] | 6 | 1 | 200 | $-\cos\theta_{1,t} - \cos(\theta_{1,t} + \theta_{2,t})$ |
| FixedSwimmer [1] | 9 | 2 | 1000 | $\dot{x}_t - 0.0001\|\mathbf{a}_t\|_2^2$ |
| Reacher [1] | 11 | 2 | 50 | $-\|\mathbf{d}_t\| - \|\mathbf{a}_t\|_2^2$ |
| Hopper [1] | 11 | 3 | 1000 | $\dot{x}_t - 0.1\|\mathbf{a}_t\|_2^2 - 3(z_t - 1.3)^2$ |
| Walker2d [1] | 17 | 6 | 1000 | $\dot{x}_t - 0.1\|\mathbf{a}_t\|_2^2 - 3(z_t - 1.3)^2$ |
| HalfCheetah [1] | 17 | 6 | 1000 | $\dot{x}_t - 0.1\|\mathbf{a}_t\|_2^2$ |
| PETS-Reacher3D [2] | 17 | 7 | 150 | $-\|\mathbf{d}_t\|_2^2 - 0.01\|\mathbf{a}_t\|_2^2$ |
| PETS-HalfCheetah [2] | 18 | 6 | 1000 | $\dot{x}_t - 0.1\|\mathbf{a}_t\|_2^2$ |
| PETS-Pusher [2] | 20 | 7 | 150 | $-1.25\|\mathbf{d}_{1,t}\|_1 - 0.5\|\mathbf{d}_{2,t}\|_1 - 0.1\|\mathbf{a_t}\|_2^2$ |
| Ant [1] | 27 | 8 | 1000 | $\dot{x}_t - 0.1\|\mathbf{a}_t\|_2^2 - 3(z_t - 0.57)^2$ |

## B  Benchmark Environment Details

In this section, we go over the details of the benchmark environments used in the experiments.

Table A.4 lists the environments along with their properties, including the dimensionality of the state $|S|$ and action spaces $|A|$, the maximum episode length. as well as the reward function. Whenever possible, we reuse the original implementations from the literature as noted in Table A.4 so as to avoid confusion. The environments that start with "PETS" are from the PETS paper [Chua et al., 2018], which is one of the baseline methods. Most other environments are from Wang et al. [2019] where the dynamics are the same as the OpenAI gym version and the reward function in Table A.4 is exposed to the agent. For more details of the environments, the readers are referred to the original paper.

Note that FixedSwimmer is a modifed version of the original Gym Swimmer environment where the velocity sensor on the neck is moved to the head. This fix was originally proposed by Wang and Ba [2020]. For the Pendulum environment, we use the OpenAI Gym version. The modified version in [Wang et al., 2019] uses a different reward function which we have found to be incorrect.

## C  Algorithms

In this section, we give the pseudo-code of the proposed algorithms DecentCEM-A and DecentCEM-P in Algorithm 1 and 2 respectively. We only show the training phase. The algorithm at inference time is simply the same process without the data saving and network update. For the internal process of CEM, we refer the readers to De Boer et al. [2005], Wang and Ba [2020].

## D  Implementation Details

### D.1  Reproducibility

Our implementation is fully reproducible by identifying the sources of randomness and controlling the random seeds as summarized in Table D.1. The seeds are set once at the beginning of the experiments.

### D.2  Hyperparameters

This section includes the details of the key hyperparameters used in the baseline algorithms *PETS* (Table D.3), *POPLIN-A/P* (Table D.4) and *SAC*[5] (Table D.2). The proposed *DecentCEM* algorithms

---

[5]Our SAC implementation used network architectures that are similar to the policy network in our method. The results of our implementation either matches or surpasses the ones reported in [Chua et al., 2018, Wang and Ba, 2020] and [Wang et al., 2019]

---
**Algorithm 1:** DecentCEM-A Training
---

1    Initialize the policy networks $p_i$ with $\theta_i, i = 1, 2, \cdots, M$ where $M$ is the ensemble size. Planning horizon $H$. Initialize the dynamics network $f_\omega$ parameterized by $\omega$. Empty Datasets $D_m$ and $D_p$

     // Episode 1, warmup phase

2    Rollout using a random policy, fill the dataset $D_m$ with the transition data $\{(s_t, a_t, s_{t+1})\}$

3    Update $\omega$ using $D_m$ by Mean-Squared Loss              // Train the dynamics network with $D_m$

     // Episode 2 onwards

4    **repeat**

5       $t = 0, D_p = \{\}$                              // Each episode, reset time and dataset

6       **repeat**

7          **foreach** *policy network $p_i$ in the ensemble* **do**

8             generate reference mean of action sequence distribution $\mu_i$ using $p_i$ and the model $f_\omega$.

               // Apply CEM to refine the action distribution.

               // $\hat{\mu}_i, v_i$ are the mean action sequence of the refined distribution and its expected value

9             $\hat{\mu}_i, v_i = \text{CEM}(\mu_i)$

10            $\hat{a}_{t,i} = \hat{\mu}_i[0]$

11          **end**

12          $a_t = \arg\max_{\hat{a}_{t,i}} v_i$                      // Pick best distribution

13          $s_{t+1} = step(a_t)$                // Execute the first action in the mean sequence

14          Append the transition $(s_t, a_t, s_{t+1})$ to $D_m$

15          Append the data $\{(s_t, \hat{a}_{t,i})\}_{i=1}^M$ to $D_p$

16          $t = t + 1$                                  // Update time step

17       **until** *Either reached the maximum episode length or terminal state*

18       Update the model parameter $\omega$ using dataset $D_m$

19       Update the policy network weights $\{\theta_i\}_{i=1}^M$ using dataset $D_p$ by the behavior cloning objective

20    **until** *bored*

---

---
**Algorithm 2:** DecentCEM-P Training
---

1    Initialize the policy networks $p_i$ with $\theta_i, i = 1, 2, \cdots, M$ where $M$ is the ensemble size. Planning horizon $H$. Initialize the dynamics network $f_\omega$ parameterized by $\omega$. Empty Datasets $D_m$ and $D_p$

     // Episode 1, warmup phase

2    Rollout using a random policy, fill the dataset $D_m$ with the transition data $\{(s_t, a_t, s_{t+1})\}$

3    Update $\omega$ using $D_m$ by Mean-Squared Loss              // Train the dynamics network with $D_m$

     // Episode 2 onwards

4    **repeat**

5       $t = 0, D_p = \{\}$                              // Each episode, reset time and dataset

6       **repeat**

7          **foreach** *policy network $p_i$ in the ensemble* **do**

               // Apply CEM to refine the distribution of the neural network weight.

               // $\hat{\mu}_i, v_i$ are the mean of the refined weight distribution sequence and its expected value

8             $\hat{\mu}_i, v_i = \text{CEM}(\theta_i)$

9             $\delta_i = \hat{\mu}_i[0]$           // Keep the weight at the first step and discard the rest

10          **end**

11          $\theta_t = \arg\max_{\theta_i + \delta_i} v_i$             // Pick the best distribution of weight sequence

12          $a_t = p_{\theta_t}(s_t)$

13          $s_{t+1} = step(a_t)$            // Execute the action returned by the policy network $p_{\theta_t}$

14          Append the transition $(s_t, a_t, s_{t+1})$ to $D_m$

15          Append the data $\{\delta_i\}_{i=1}^M$ to $D_p$

16          $t = t + 1$                                  // Update time step

17       **until** *either reached the maximum episode length or terminal state*

18       Update the model parameter $\omega$ using dataset $D_m$

19       Update the policy network weights $\{\theta_i\}_{i=1}^M$ using dataset $D_p$ by the AVG training objective

20    **until** *bored*

---

Table D.1: Random Seed. The set {1,2,3,4,5} refers to the seeds for five runs. Note that we control the random seed for the environments since there is a random number generator in openai gym environments independent from other sources

| Source of randomness | Random Seed |
|---|---|
| Tensorflow | |
| numpy | {1,2,3,4,5} |
| python random module | |
| the training environment | 1234 |
| the evaluation environment | 0 |

Table D.2: Hyperparameters of SAC

| Parameter | Value |
|---|---|
| Actor learning rate | 0.0001 |
| Critic learning rate | 0.0001 |
| Actor network architecture | $[|S|, 64, 64, 2\times |A|]$ |
| Critic network architecture | $[|S| + |A|, 64, 64, 1]$ |

Table D.3: Hyperparameters of PETS

| Parameter | Value |
|---|---|
| Model learning rate | 0.001 |
| Warmup episodes | 1 |
| Planning Horizon | 30 |
| CEM population size | 500 (400 in PETS-reacher3D) |
| CEM proportion of elites | 10% |
| CEM initial distribution variance | 0.25 |
| CEM max # of internal iterations | 5 |

Table D.4: Hyperparameters of POPLINA and POPLINP

| Parameter | Value |
|---|---|
| Model learning rate | 0.001 |
| Warmup episodes | 1 |
| Planning Horizon | 30 |
| CEM population size | 500 (400 in PETS-reacher3D) |
| CEM proportion of elites | 10% |
| CEM initial distribution variance | 0.25 |
| CEM max # of internal iterations | 5 |
| Policy network architecture (A) | $[|S|, 64, 64, |A|]$ |
| Policy network architecture (P) | $[|S|, 32, |A|]$ |
| Policy network learning rate | 0.001 |
| Policy network activation function | tanh |

(*DecentPETS, DecentCEM-A, DecentCEM-P*) have identical hyperparameters as their corresponding baselines (*PETS, POPLIN-A, POPLIN-P*) except for an additional ensemble size parameter. The hyperparameter search for the ensemble size is performed by sweeping through the set $\{2, 3, 4, 5, 6\}$ for each environment. For the neural network architecture for the dynamics model, the *DecentCEM* methods exactly follow the original one in *PETS* and *POPLIN* for a fair comparison, which is an ensemble of fully connected networks.

# E  Full Results

## E.1  Detailed visualization of the iterative updates in the one-dimensional optimization task

Figure 10 is a version of Figure 4 with more iterations. It shows the iterative sampling process of CEM methods in the 1D optimization task and how the sampling distributions evolve over time.

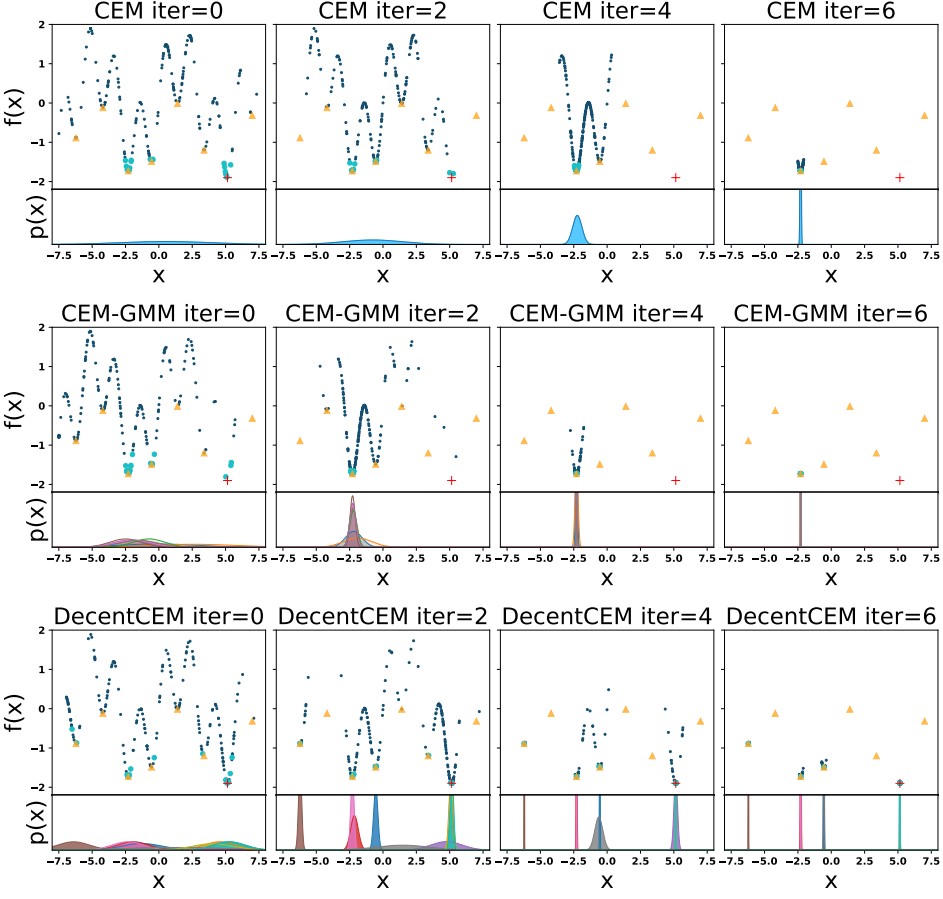

Figure 10: The iterative sampling process in the one-dimensional optimization task.

## E.2 Full Learning Curves

In Figure 11, we report the learning curves in all environments listed in Appendix B.

The algorithms evaluated in the benchmark are: PETS, POPLIN-A, POPLIN-P and the proposed methods *DecentPETS, DecentCEM-A* and *DecentCEM-P*. We also included a model-free algorithm SAC as a baseline. *DecentCEM* subsumes POPLIN and they are equivalent when the ensemble size is one. The same applies to *DecentPETS* and PETS. To distinguish them in the learning curves and discussions, we show the *DecentCEM* results from an ensemble size larger than one.

The learning curves in some environments can be noisy. We apply smoothing with 1D uniform filter. The window size of the filter was 10 for all but Cartpole, where 30 was used due to its large noise for all algorithms.

Note that the performance of the baseline methods may be different from the results reported in their original paper. Specifically, in the paper by Wang and Ba [2020], PETS, POPLIN-A and POPLIN-P have been evaluated in a number of environments that we use for the benchmark. Our benchmark results may not be consistent with theirs due to differences in the implementation and evaluation protocol. For example, our results of PETS, POPLIN-A and POPLIN-P in the Acrobot environment are all better than the results in Wang and Ba [2020]. We have identified a bug in the POPLIN code base that causes the evaluation results to be on a wrong timescale that is much slower than what it actually is. Hence the results of our implementation look far better, reaching a return of 0 at about 4k steps as opposed to 20k steps reported in Wang and Ba [2020].

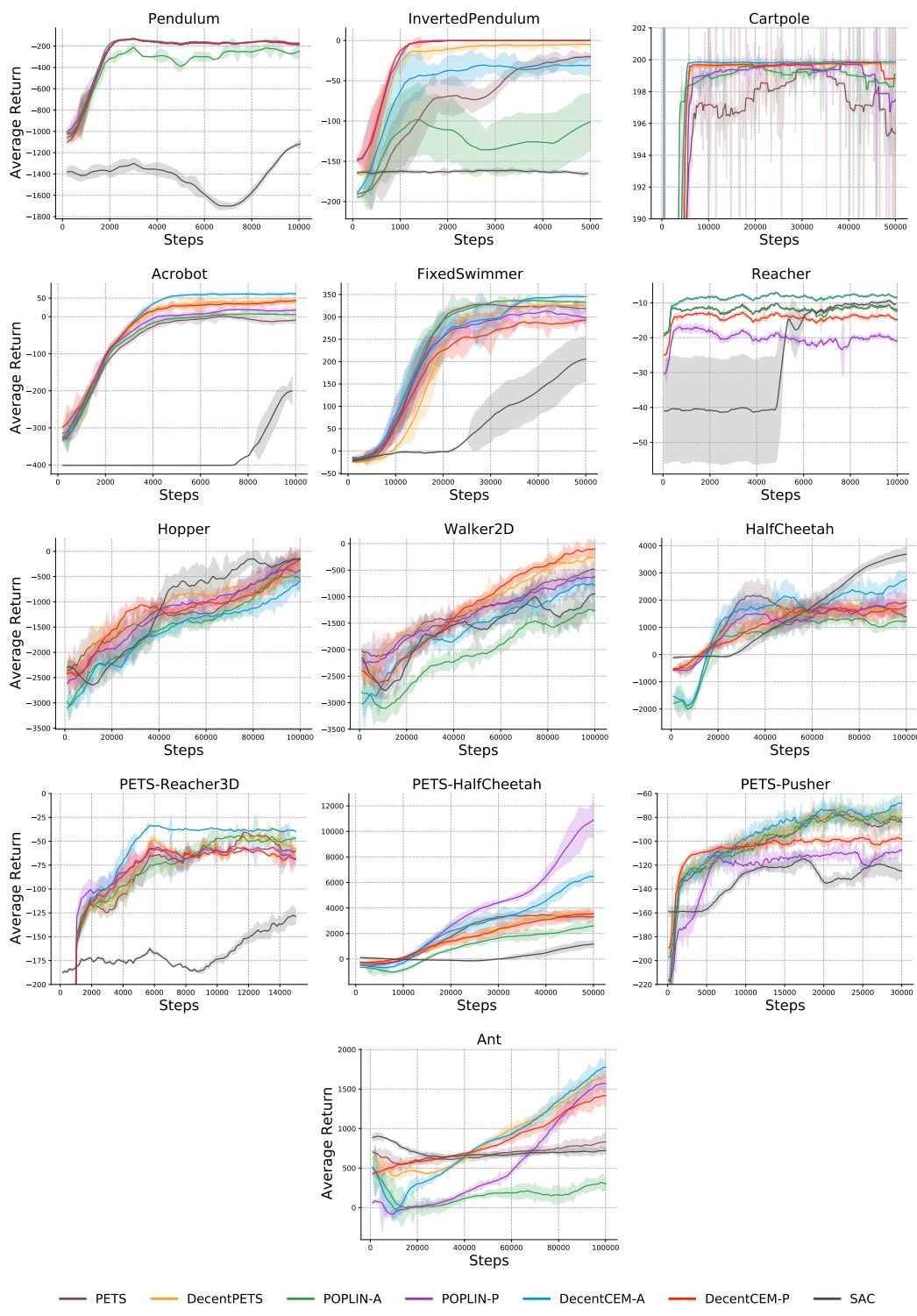

Figure 11: The learning curves of the proposed *DecentCEM* methods and the baseline methods on continuous control environments. The line and shaded region shows the mean and standard error of evaluation results from 5 training runs using different random seeds. Each run is evaluated in an environment independent from training and reports average return of 5 episodes at every training episode. To ensure that the evaluation environments are the same across different methods and multiple runs, we set a fixed random seed in the evaluation environment of each task.

### E.3 Analysis

Let's group the environments into two categories based on how well the *decentralized* methods perform in them:

1. Pendulum, InvertedPendulum, Acrobot, Cartpole, FixedSwimmer, Reacher, Walker2D, PETS-Pusher, PETS-Reacher3D, Ant
2. Hopper, HalfCheetah, PETS-HalfCheetah

The first category is where the best performing method is one of the proposed *decentralized* algorithms: *DecentPETS*, *DecentCEM-A* or *DecentCEM-P*. In environments where the baseline *PETS*, *POPLIN-A* or *POPLIN-P* could reach near-optimal performance such as pendulum and invertedPendulum, applying the ensemble method would yield similar performance as before. It is also evident from InvertedPendulum results that the decentralized version significantly improves over the centralized algorithm where the performance of the latter is poor. In Cartpole, Acrobot, Reacher, Walker2D and PETS-Pusher, applying the decentralized approach increases the performance in both action space planning ("A") and parameter space planning ("P"). In Pendulum, InvertedPendulum, FixedSwimmer, PETS-Reacher3D and Ant, ensemble helps in the action space planning but either has no impact or negative impact on the parameter space planning.

The second category is where it is better not to use a *decentralized* approach with multiple instances (note that the decentralized methods with one instance fall back to one of *PETS*, *POPLIN-A*, *POPLIN-P*). In Hopper and HalfCheetah, the issue might lie in the model rather than planning since all MBRL baselines performed worse than the model-free baseline SAC. In HalfCheetah, *DecentCEM-A* in fact performs the best out of all model-based methods but it falls behind SAC. One possible issue is that the true dynamics is difficult to approximate with our model learning approach. Another possibility is that it may be necessary to learn the variance of the sampling distribution, which none of these model-based approaches do. To be clear, the variance is *adapted* online by CEM but it is not *learned*. PETS-HalfCheetah is slightly different in that the ensemble does improve the performance significantly when used for action space planning. However, *POPLIN-P* performs significantly better than all other algorithms. This suggests that the parameter space planning has been able to successfully find a high return region using a single Gaussian distribution. In this case, distributing the population size would not be able to trade the estimation accuracy for a better global search.

One interesting phenomenon is that DecentPETS performs better than or comparably as PETS in *all* environments in both categories except in Hopper (where they are quite close as well). This suggests that when not using a learned neural network to initialize the distribution in CEM, decentralizing the samples is an effective technique to achieve an improvement of the optimization performance.

## F   More Ablations

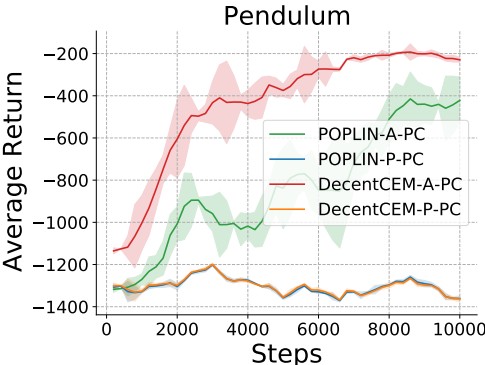

Figure 12: Policy control ablation where the policy network is directly used for control without CEM policy improvement

We study the performance of policy control where the policy network is directly used for control without the CEM step, denoted by the extra suffix "-PC". The result is shown in Fig. 12. Without the

policy improvement from CEM, all algorithms perform worse than their counter-part of using CEM. *POPLIN-P-PC* and *DecentCEM-P-PC* both get stuck in local optima and do not perform very well. This makes sense since the premise of planning in parameter space is that CEM can search more efficiently there. The policy network is not designed to be used directly as a policy. Interestingly, *DecentCEM-A-PC* achieves a high performance from about 7k steps (35 episodes) of training. The ensemble of policy networks seems to add more robustness to control compared to using a single one.

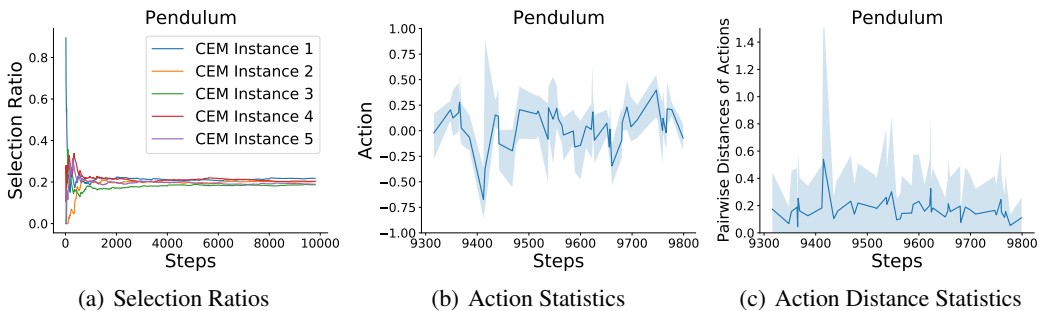

(a) Selection Ratios       (b) Action Statistics       (c) Action Distance Statistics

Figure 13: Ablation of ensemble diversity in Pendulum during training of *DecentPETS* with 5 instances. (a) Cumulative selection ratio of each CEM instance. (b)(c) Statistics of the actions and pairwise action distances of the instances, respectively. The line and shaded region represent the mean and min/max.

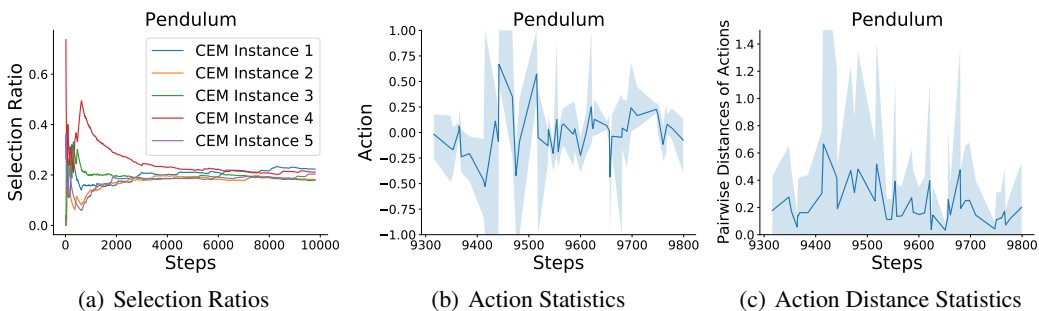

(a) Selection Ratios       (b) Action Statistics       (c) Action Distance Statistics

Figure 14: Ablation of ensemble diversity in Pendulum during training of *DecentCEM-P* with 5 instances. (a) Cumulative selection ratio of each CEM instance. (b)(c) Statistics of the actions and pairwise action distances of the instances, respectively. The line and shaded region represent the mean and min/max.

Figure 13 and 14 are additional plots for the ensemble diversity ablation. They show the results for DecentPETS and DecentCEM-P, respectively. The same as in Fig. 9 (b)(c), we only show a time window toward the end of training for visual clarity and the line and shaded region represent the mean and min/max. Comparing the action and action distances statistics of the three algorithms shown in Fig. 9 (b)(c), 13 (b)(c), 14 (b)(c), the actions from *DecentPETS* cover a smaller range of values compared with those from *DecentCEM-A/P*. This suggests that the use of policy networks in the multiple instances add more exploration without sacrificing performance thanks to the $\arg\max$.

# G  Overhead of the Ensemble

The sample efficiency is not impaired when going from one policy network to the multiple policy networks used in *DecentCEM-A* and *DecentCEM-P*. This is because that the generation of the training data only involve taking imaginary rollouts with the model, rather than interacting with the real environment, as discussed in Section 4.

In terms of the population size (number of samples drawn in CEM), the *DecentCEM* methods do not impose additional cost. We show in both the motivational example (Sec. 3) and the benchmark experiments (Appendix E) that the proposed methods work better than *CEM* under the same total population size.

The additional computational cost is reasonable in *DecentCEM* compared to *POPLIN*. Each branch of policy network and CEM instance runs independently from the others, allowing for a parallel implementation. The instances have to be synchronized ($\arg\max$) but its additional cost is minimal. One caveat with our current implementation though is that the instances run serially, which slows down the speed. This is not a limitation of the method itself though and the speed loss can be alleviated by a parallel implementation.

# H  Convergence Analysis of Decentralized CEM

This section analyzes the convergence of the proposed DecentCEM algorithm in optimization.

Consider the following optimization problem:

$$x^* \in \arg\max_{x \in \mathcal{X}} V(x) \tag{4}$$

where $\mathcal{X} \subset \mathbb{R}^n$ is a non-empty compact set and $V(\cdot)$ is a bounded, deterministic value function to be maximized. We assume that this problem has a unique global optimal solution $x^*$ but the objective function $V(\cdot)$ may have multiple local optimum and may not be continuous.

We will show that the existing convergence result of CEM in continuous optimization established in [Hu et al., 2011] also applies to DecentCEM. It assumes that the sampling distribution $g_\phi(x)$ in CEM is in the natural exponential families (NEFs) which subsumes Gaussian distribution (with known covariance). We restate the definition of NEFs for completeness (definition 2.1 in [Hu et al., 2011]):

**Definition H.1** (Natural Exponential Family). *A family of parameterized distributions $\{g_\phi(\cdot), \phi \in \Phi \subset \mathbb{R}^d\}$ on $\mathcal{X} \subset \mathbb{R}^n$ is called a Natural Exponential Family (NEF) if there exists continuous mappings $\Gamma : \mathbb{R}^n \to \mathbb{R}^d$, $h : \mathbb{R}^n \to \mathbb{R}$ and $K : \mathbb{R}^d \to \mathbb{R}$ such that $g_\phi(x) = exp(\phi^\top \Gamma(x) - K(\phi))h(x)$, where the parameter space $\Phi = \{\phi \in \mathbb{R}^d : |K(\phi)| < \infty\}$, $K(\phi) = \ln \int_{\mathcal{X}} exp(\phi^\top \Gamma(x))h(x)\nu(dx)$ and $\nu$ is the Lebesgue measure of $\mathcal{X}$.*

The mean vector function

$$m(\phi) = \mathbb{E}_\phi[\Gamma(x)] \tag{5}$$

where the expectation $\mathbb{E}_\phi$ is with respect to $g_\phi$ and $\Gamma$ is the mapping in Def. H.1. It can be shown that $m(\cdot)$ is invertible. Note that the expression of the densities can be simplified when restricted to a multivariate Gaussian distribution (with known diagonal covariance) where the natural sufficient statistics $\Gamma(x) = x$.

We then present the CEM algorithm below to fix notations. It follows Algorithm 2 in [Hu et al., 2011] but is modified to align with some notations introduced in previous sections in our paper.

The convergence results will require the following assumptions from Hu et al. [2011]:

**Assumption 2.** *The parameter $\phi_{k+1}$ computed at step 3 of Algorithm 3 satisfies $\phi_{k+1} \in int(\Phi)$ for all $k$.*

**Assumption 3.** *The step size sequence $\{\alpha_k\}$ satisfies: $\alpha_k > 0 \,\forall\, k$, $\lim_{k \to \infty} \alpha_k = 0$ and $\sum_{k=0}^{\infty} \alpha_k = \infty$.*

**Assumption 4.** *The sequence $\{\lambda_k\}$ satisfies $\lambda_k = O(k^{-\lambda})$ for some constant $\lambda \geq 0$ and the sample size $N_k = \Theta(k^\beta)$ where $\beta > \max\{0, 1 - 2\lambda\}$.*

**Assumption 5.** *The $(1 - \rho)$-quantile of $\{V(x), x \sim g_\phi(x)\}$ is unique for each $\phi \in \Phi$.*

We know from Hu et al. [2011] that the sequence $\{\eta_k\}_{k=0}^{\infty}$ from equation 7 asymptotically approaches the solution set of the ODE:

$$\frac{d\eta(t)}{dt} = L(\eta) \tag{8}$$

$$L(\eta) = \nabla_\phi \ln \mathbb{E}_\phi[\mathbb{1}(V(x), \gamma(m^{-1}(\eta)))]\,|_{\phi=m^{-1}(\eta)} \tag{9}$$

where $\gamma(m^{-1}(\eta))$ is the true $(1 - \rho)$-quantile of $V(x)$ under $g_{m^{-1}(\eta)}$.

**Assumption 6.** *The function $L(\eta)$ defined in equation 9 has a unique integral curve for a given initial condition.*

---

**Algorithm 3:** CEM

---

1  Choose the family of distributions $g_\phi(x), x \in \mathcal{X}$ from NEFs defined in H.1 and the initial parameter
   $\phi_0 \in int(\Phi)$ where $int$ denotes the interior of the parameter space $\Phi$.
2  Specify elite ratio $\rho \in (0, 1)$ and step size sequence $\{\alpha_k\}$ and $\{\lambda_k\}$ where $k$ denotes the time step. Set
   $k = 0$. Specify $\epsilon > 0$ which is the parameter in the thresholding function defined in Equation 3.
3

$$\mathbb{1}(x, \gamma) = \begin{cases} 1, & \text{if } x \geq \gamma \\ \frac{x - \gamma + \epsilon}{\epsilon}, & \text{if } \gamma - \epsilon < x < 1 \\ 0, & \text{if } x \leq \gamma - \epsilon \end{cases} \tag{6}$$

4  **repeat**
5     Step 1: Draw $N_k$ i.i.d samples $\Lambda_k = \{x_1, x_2, ..., x_{N_k}\}$ from the distribution $g_{\phi_k}(x)$
6     Step 2: Calculate the sample $(1 - \rho)$-quantile $\hat{\gamma}_k = V_{(\lceil (1-\rho)N_k \rceil)}$ where $\lceil a \rceil$ is the ceiling function that
      gives the smallest integer greater than $a$ and $V_{(i)}$ is the $i$th-order statistics of the sequence
      $\{V(x_j)\}_{j=1}^{N_k}$ where $V(\cdot)$ is the objective function to be maximized.
7     Step 3: Compute the new parameter $\phi_{k+1} = m^{-1}(\eta_{k+1})$, where $\eta_0 = m(\phi_0) = \mathbb{E}_{\phi_0}(\Gamma(x))$ and

$$\eta_{k+1} = \alpha_k \frac{\sum_{x \in \Lambda_k} \mathbb{1}(V(x), \hat{\gamma}_k)\Gamma(x)}{\sum_{x \in \Lambda_k} \mathbb{1}(V(x), \hat{\gamma}_k)} +$$

$$(1 - \alpha_k)\left(\frac{\lambda_k}{N_k} \sum_{x \in \Lambda_k} \Gamma(x) + (1 - \lambda_k)\eta_k\right) \tag{7}$$

8     Step 4: $k = k + 1$
9  **until** *a stopping condition is reached*
10 **return** $\phi_k$

---

The above assumptions 2-6 are the assumptions required by the previous convergence result of CEM. To show the convergence of DecentCEM, we only require one additional mild condition in the assumption 1 (note that the sample size requirement is included here only for completion since it is already part of Assumption 4).

Now we restate the convergence result of DecentCEM from the main text and show the proof:

**Theorem 3.1** (Convergence of DecentCEM). *If a CEM instance described in Algorithm 3 converges, and we decentralize it by evenly dividing its sample size $N_k$ into $M$ CEM instances which satisfies the Assumption 1, then the resulting DecentCEM converges almost surely to the best solution of the individual instances.*

*Proof.* Each individual CEM instance has a sample size of $\frac{N_k}{M}$ and $N_k = \Theta(k^\beta)$. Since Assumption 1 holds, $M$ is constant and gets absorbed into the $\Theta$ and we have $\frac{N_k}{M} = \Theta(k^\beta)$. Hence the conditions of Theorem 3.1 in [Hu et al., 2011] holds for each CEM instance indexed by $i$ and can be directly applied to show the almost sure convergence of their solutions $\{\eta_{i,k}\}$ to an internally chain recurrent set of Equation 8. If the recurrent sets are isolated equilibrium points, then $\{\eta_{i,k}\}$ converges almost surely to a unique equilibrium point.

Due to the fact that the instances in DecentCEM run independently from each other, their solutions $\{\eta_{i,k}\}_{i=1}^M$ (or equivalently $\{\phi_{i,k}\}_{i=1}^M = \{m^{-1}(\eta_{i,k})\}_{i=1}^M$) might converge to identical or different solutions denoted as $\{\eta_i^*\}_{i=1}^M$. DecentCEM computes the final solution by applying an $\arg\max$ over all individual solutions: $\eta_{o,k} = \arg\max_{\eta \in \{\eta_{i,k}\}_{i=1}^M} \mathbb{E}_{m^{-1}(\eta)}[V(x)]$ (equivalent to Equation 3). Here the expectation is approximated by the sample mean with respect to the distribution $g_{m^{-1}(\eta)}$: $\frac{1}{N_k} \sum_{j=1}^{N_k} V(x_j)$, which converges almost surely to the true expectation according to the strong law of large numbers. Hence we have that $\eta_{o,k}$ converges almost surely to the best solution in the set $\{\eta_i^*\}_{i=1}^M$ found by the individual CEM instances, in terms of the expected value of $\mathbb{E}_{m^{-1}(\eta)}[V(x)]$. $\square$

Note that the theorem implies that the solution of CEM / DecentCEM assigns the maximum probability to a locally optimal solution to Equation 4. It does not guarantee whether this local optimum is a global optimum or not. To the best of our knowledge, almost sure convergence to a local optimum is the only convergence result that has been established about CEM in continuous optimization.

# I  Visualization of DecentCEM Planning

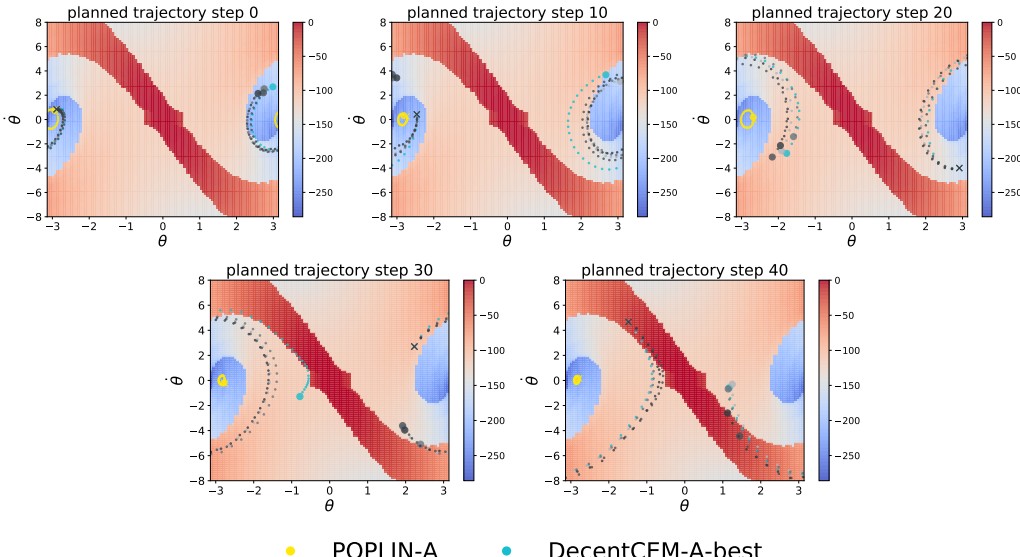

Figure 15: Planning trajectories of DecentCEM-A (ensemble size 5) and POPLIN-A. It's overlaid on the heatmap showing the true optimal state value (red means higher value). Title of each plot shows the steps in the evaluation environment. Each planned trajectory has 30 steps, denoted by a sequence of dots. The starting state is denoted by a cross $\times$ and the ending (planned) state is denoted by a big solid dot ● (with different colors corresponding to different trajectories). The color for POPLIN-A and the best solution from DecentCEM-A is indicated in the legend in the plot and the rest of DecentCEM-A trajectories are colored with different shades of gray. Note that $\theta$ of $2\pi$ and $-2\pi$ are identical in the angular position but appears as "disconnected" on the plots.

To better understand the planning process of DecentCEM, we visualized the planning trajectories of POPLIN-A and DecentCEM-A in Fig.15. The planned state trajectories are denoted by sequences of dots. Each plot show the planned trajectories at different steps from running both algorithms (at 2k training steps) on the same evaluation environment such that the comparison is fair. The heatmap shows the optimal state value solved by value iteration on the discretized pendulum environment. The discretization is performed by discretizing the state space and action space of the original pendulum environment into 100 and 50 intervals respectively. The best trajectory from the multiple instances in DecentCEM-A is colored in cyan. Note that it is not ranked based on the true value, but on the imaginary value during planning. We could observe that this solution may not always be the true best solution among the trajectories due to the model inaccuracy at 2000 training steps. However, these trajectories are able to explore the space better than using a single instance as in POPLIN-A which can easily get stuck in the state regions with low values.