# OpenReview forum: "A Simple Decentralized Cross-Entropy Method"
_NeurIPS.cc/2022/Conference — NeurIPS 2022 Accept_

### Official Review · Reviewer_cLuK · 2022-07-11

**Rating:** 6
**Confidence:** 3
**Soundness:** 3 good
**Presentation:** 2 fair
**Contribution:** 1 poor

**Summary:**

This paper proposes DecentCEM, where two or more instances of cross-entropy method (CEM) are run independently with different sets of samples until convergence. For model-based decision making, the results from the ensemble are gathered and combined by choosing the best sampling distribution that maximizes the expected return of trajectories. Theoretically, they show that each instance of CEM converges under the same set of assumptions for convergence of vanilla CEM. They combine DecentCEM with CEM planning and empirically show that DecentCEM outperforms baseline methods including PETS and POPLIN in multiple environments for MBRL.

**Questions:**

- I would like to check out the details and a more thorough version of Table A.3. How many instances or mixture components were employed for the table?
- Also, please see "Strengths And Weaknesses" for additional relevant comments.

**Limitations:**

I think one main limitation of the method is fairly covered, regarding the discrepancy between the theoretical result and practice.

**Strengths And Weaknesses:**

Originality

- In terms of the originality, the idea of running multiple instances of the same algorithm and combining the results is not very novel, as it is one of the common patterns in problem solving.

Quality

- The method is sound, and I agree that it could bring improvements in multiple situations.
- I think Theorem 3.1 (Convergence of DecentCEM) needs to be more specific about the "almost-sure" convergence of the solution of DecentCEM. Even if they maintain the expression "almost surely", at least they should state that it holds for large-enough sample sizes.

Clarity

- Overall, the manuscript is easy to follow and well-organized. Especially, the early introduction of the motivating examples can help better understanding of the paper.
- Some parts of the paper could mislead readers and possibly oversell the results of the paper. For instance, in the abstract, the authors mention that they "provide both theoretical and empirical analysis" and the next sentence states that they "show" that "DecentCEM finds the global optimum much more consistently". This could easily make readers think that such statement is backed by theoretical results as well, which is not true according to my understanding.
- Regarding Theorem 3.1 (Convergence of DecentCEM), please see my comment above on quality.

Significance

- Even though the proposed decentralized CEM is not fully novel algorithmically, suggesting that such approach yields valid results with potential improvements can still be somewhat meaningful.

---

> ### Author Response · Authors · 2022-08-02
> **Response to Reviewer cLuK**
>
> Dear reviewer:
>
> Thank you for your time reviewing our paper and your thoughtful feedback.
> We have uploaded the revised main paper and appendix (all changes marked in blue color) to incorporate your feedback.
> Below please find our responses to your questions and comments:
>
> > state that it (the theorem) holds for large-enough sample size
>
> `Re:` in the revision, we have restructured the theorem where we state the assumption of sample size prior to introducing the theorem.  Hopefully that makes it clearer.
>
> > clarify in abstract
>
> `Re:` we addressed it in the revision.
>
> > the details and a more thorough version of Table A.3. How many instances or mixture components were employed for the table?
>
> `Re:`The results in Table A.3 are based on the best hyper-parameters in Table A.2 (explained in the paragraph in A.1). E.g., the total time for CEM-GMM and DecentCEM under sample size of 100 would be based on the hyperparameters in the  column corresponding to sample size 100 in Table A.2: hence 10 mixture components for GMM and 10 instances for DecentCEM. Similarly, for sample size 200,  it’s 8 mixture components for GMM and 10 instances for DecentCEM.

---

> > ### Comment · Reviewer_cLuK · 2022-08-09
> > **Response to Authors**
> >
> > I appreciate the authors for the response!
> >
> > The authors addressed my concerns fairly, and I'm raising my score to 6.

---

> > > ### Author Response · Authors · 2022-08-10
> > > **Thank you for your response**
> > >
> > > Dear Reviewer,
> > >
> > > Thank you for reading our rebuttal and for raising the score.

---

### Official Review · Reviewer_H4mJ · 2022-07-11

**Rating:** 5
**Confidence:** 4
**Soundness:** 3 good
**Presentation:** 2 fair
**Contribution:** 2 fair

**Summary:**

This paper propose a multi-instance CEM method, a strict generalization of CEM, which is able to remain spread-out in the search space. The authors test on continuous-action model-based RL.


**Questions:**

Footnote 2 seems incomplete.

How is BiggerPolicyNet tuned in the ablation study?

How can DecenCEM be used in settings other than continuous-action MBRL?

How does the performance compare to gradient based methods?


**Limitations:**

+ The authors are upfront about their setting (continuous action, requires a model, etc.).


**Strengths And Weaknesses:**

The method uses common optimization philosophy and have shown good performance; and the analysis is sound. The paper is overall well written and have included an example for intuition.

The method design principled and given theoretical attention.

However, the empirical evaluation hardly distinguish the method from the baselines, that is, the performance gain appears to be minimal.

The novelty of the proposed method is not very significant; however, it can be considered ground work for many future works.

---

> ### Author Response · Authors · 2022-08-02
> **Response to Reviewer H4mJ**
>
> Dear Reviewer:
>
> Thank you for your time reviewing our paper and your thoughtful feedback.
> Below please find our responses to your questions and comments:
>
> We have incorporated your writing suggestions into the revised manuscript.
>
> > Footnote 2 seems incomplete
>
> `Re`: It is complete as to what we wanted to put there.
>   Perhaps the confusion was led by the way that the last sentence was phrased. We addressed it in the revision.
>
> > How is biggerpolicynet turned in the ablation study
>
> `Re:`  Each NN layer in the policy net as the same number of nodes (except the input and output layer).
> We simply added more intermediate layers to the original policy net and choose the number of layers such that the total number of NN  parameters are closest to the sum of all params in the ensemble.
>
> > How can DecentCEM be used in settings other than continuous-action MBRL.
>
> `Re:`
> The seems to be two aspects to this question.
>
> 1. As to discrete-action setting: it’s trivial to make CEM work with discrete action space (just need to use discrete distributions). Theory-wise, it is easier to show the convergence of CEM under discrete support than a continuous one.
> 2.  As to non MBRL setting: since CEM is an optimization algorithm, it can be and has been applied to Model-free RL settings.
> A notable example is the Qt-opt algorithm behind the robotics at the google robotics farm where the argmax of the Q function is performed by CEM.
>
> Thus DecentCEM can be easily adapted to the above two scenarios.
>
> > How does the performance compare to gradient based methods
>
> `Re:`  Some comparison of gradien-based and PETS (CEM-based) has been performed by Wang et al 2019 (in our reference).
> To have such a comparison with DecentCEM would require rigorously re-running gradient-based methods under our setting. Our evaluation protocol is more rigorous than the one in [Wang et al 2019] (they report training performance rather than evaluation) hence not directly comparable. 1 data point in the learning curve is from 5(train) runs in their paper and 5(train)*5(test)=25 runs in ours.
> We think such benchmark experiments are beyond the scope of our paper and would leave it for future work.

---

> > ### Comment · Reviewer_H4mJ · 2022-08-08
> > **Response**
> >
> > Thank you for the clarification, I have raised my score.

---

> > > ### Author Response · Authors · 2022-08-08
> > > **Thank you for your response**
> > >
> > > Dear Reviewer,
> > >
> > > Thank you for reading our rebuttal and for raising the score.

---

### Official Review · Reviewer_PaJC · 2022-07-12

**Rating:** 6
**Confidence:** 4
**Soundness:** 3 good
**Presentation:** 2 fair
**Contribution:** 3 good

**Summary:**

Cross-Entropy Method (CEM) is commonly used for planning in model-based reinforcement learning (MBRL) where the *centralized* approach is typically utilized to update the sampling distribution based on only the top-*k*  operations' results on samples. In this paper, the authors show that such a *centralized* approach makes CEM vulnerable to local optima, thus impairing its sample efficiency. To tackle this issue,  the authors propose **Decent**ralized **CEM (DecentCEM)**, a simple but effective improvement over classical CEM, by using an ensemble of CEM instances running independently from one another, and each performing a local improvement of its own sampling distribution. The theoretical and empirical analysis are provided to demonstrate the effectiveness of this simple *decentralized* approach. It is shown that compared to the classical centralized approach using either a single or even a mixture of Gaussian distributions, their DecentCEM finds the global optimum much more consistently thus improving the sample efficiency. Furthermore, they plug in their DecentCEM in the planning problem of MBRL and evaluate their approach in several continuous control environments, with comparison to the state-of-art CEM-based MBRL approaches (PETS and POPLIN). Results show sample efficiency improvement by simply replacing the classical CEM module with their DecentCEM module, while only sacrificing a reasonable amount of computational cost, which can be alleviated by using the advantage of the parallelism of their ensemble method. Lastly, they conduct ablation studies for more in-depth analysis.

**Questions:**

1. Could you please add **DescentPETS** as the baseline to make the experiment more comprehensive? **DescentPETS** means replacing the CEM module in the previous PETS algorithm with the DescentCEM proposed in this paper.

2. Could you please investigate the effectiveness of the **DecentCEM** both theoretically and empirically when assumption 2 is alleviated?

3. Could you please visualize the planning trajectories with **CEM** and *DecentCEM** and compare them if possible?

**Limitations:**

1. We can notice that the theoretical convergence result does not match the empirical observation well. Please investigate this deeply and explore more effective methods to improve the algorithm's practical convergence result.

2. Could you please investigate the theoretical convergence result of your proposed algorithm under the *constant sample size* constraint as what you have used in your experiment?

3. Could you please explore how to promote your method to other non-CEM-based MBRL algorithms?

**Strengths And Weaknesses:**

**Strengths**
1. The proposed is simple but effective. This work can be utilized in any CEM-based model-based reinforcement learning algorithm.
2. The paper is well organized and the writing is good.
3. The theoretical and empirical analysis are persuasive to demonstrate the effectiveness of the proposed method.


**Weaknesses**
1.  The assumptions of the proposed algorithm are a little strong, especially the second point. The authors assume that the reward function is known and deterministic which is not common in practice actually.

2. The visualization part in the experiment still needs to be improved.

---

> ### Author Response · Authors · 2022-08-02
> **Response to Reviewer PaJC**
>
> Dear Reviewer:
>
> Thank you for your time reviewing our paper and your thoughtful feedback.
> The main questions were on adding additional results on DecentPETS and visualization of the planning trajectories.
> We added both in the revised main paper and appendix (changes marked in blue color).
> Specifically,
> 1. We added DecentPETS results to Fig. 6 with an ensemble setup of DecentPETS following that of DecentCEM-A.
> 2. We added a section at the end of the appendix for the visualization of planning trajectories of DecentCEM-A and POPLIN-A.
>
> Below please find our responses to other questions and comments:
>
> > `assumptions are a little strong…. the reward function is known and deterministic which is not common in practice actually `
>
> `Re:` It is true that not all papers adopted this assumption. But the baselines that we compared against do use such assumptions so we followed such a setting. We cited [Agarwal et al. 2019] on why such assumptions are mild. Basically, all Mujoco environments use the deterministic reward function to the best of our knowledge, and the reward function is typically not too hard to learn. It’s certainly ideal to lift this assumption but we feel that it’s beyond the current scope of the paper, and leave it for future work to learn the reward function.
>
> > Limitations
>
> `Re:` For the limitations that you listed, thanks for the suggestions and we will study them in future work. We wanted to note that the constant sample size underlies a fundamental theory-practice gap in CEM research (not specific to the decentralized version) as we described in the future work section.  To investigate DecentCEM performance under such a setting, we’d need advancements in vanilla CEM convergence results first, which we feel to be beyond the scope of the paper. It's certainly an interesting topic to pursue down the road though.
>
> We hope that we have sufficiently addressed your concerns. We would be happy to answer any remaining questions that you might have.

---

> > ### Comment · Reviewer_PaJC · 2022-08-09
> > **Response to the authors**
> >
> > Thanks for your response. I am sorry you have not addressed my concerns well. First, though the reward function is deterministic in most Mujoco environments, reinforcement learning is not only an algorithm targeting Mujoco-like simple environments. Actually, there are many non-deterministic reward function settings in real applications. Such assumptions strongly limit the future and application of the proposed method. Second, I do not agree that the reward function is typically not too hard to learn. You will find that things do not go so smoothly if you try it on some Atari Games which are also a common benchmark for reinforcement learning research. Third, I am a little disappointed to see that your work has not brought some theoretical breakthroughs from CEM research which also limits the impact and the future of your work.
> > So I will not improve the rating score, though this is still a simple but effective work.

---

> > > ### Author Response · Authors · 2022-08-10
> > > **Response to post-rebuttal feedback**
> > >
> > > Dear Reviewer,
> > >
> > > Thank you for reading our rebuttal and for providing further feedback.
> > >
> > > `Re the reward function`: we agree that there are "many non-deterministic reward function settings in real applications" and that learning reward functions in Atari can be difficult. We wanted to point out that the proposed approach, as well as the CEM baselines, are not fundamentally limited to known deterministic reward functions (e.g., the dynamics neural network is probabilistic, line 196-198. The same technique can be used to learn a stochastic reward function). The experiments from these baseline papers, however, were conducted on MuJoCo envs (deterministic) and with known reward functions. We hence limited our scope to this same setting and made such assumptions explicit upfront (not always the case in the literature). This was necessary to enable effective comparison with existing baselines.  Footnote 2 is further revised accordingly (in the latest paper revision) to make this clear. We hope that we are not penalized for doing so.
> > >
> > > `Additional experiments`: we would like to highlight that we have added additional experiments according to your suggestions in the previous round. This was a nontrivial effort, but we are glad that they are now included and that they support our main points.  We hope that these additions are satisfactory.

---

### Official Review · Reviewer_XxHs · 2022-07-16

**Rating:** 6
**Confidence:** 3
**Soundness:** 3 good
**Presentation:** 1 poor
**Contribution:** 3 good

**Summary:**

This paper proposed a decentralised version of the cross-entropy method for model-based reinforcement learning. The convergence of the proposed algorithm is analysed under several assumptions (Hu et al. [2011]) and evaluated in several continuous control environments compared with PETS and POPLIN.

**Questions:**

1. Where is the definition of the initial sampling distribution $g_{\phi}(\tau)$? The first time it appears is in line 91. It is difficult for readers unfamiliar with planning (MPC) to proceed.

2. What is $d_a$ in line 93?

3. Theorem 3.1 is entirely broken. Where are all the assumptions? It is not pleasant to just say, “The detailed proof is left to Appendix H”.


**Strengths And Weaknesses:**

**Strengths**
1.  Decentralization of CEM in MBRL can deviate from local optima and improve sample efficiency over its centralisation counterpart.
2. A relatively complete theoretical and empirical analysis and in-depth ablation studies.
3. Evaluating planning tasks in different continuous control environments are nicely done.

**Weakness: **
The readability is impossible without referring appendix with many cross-reference links broken.

---

> ### Author Response · Authors · 2022-08-02
> **Response to Reviewer XxHs**
>
> Dear reviewer:
>
> Thank you for your time reviewing our paper and your thoughtful feedback.
> We have uploaded the revised main paper and appendix (changes marked in blue color) to incorporate your feedback.
>
> Below please find our responses to your questions and comments:
>
> > Broken cross-reference links.
>
> `Re:` We have included the main paper in the revised appendix pdf to fix this issue.
>
> > $g_\phi$
>
> `Re:` it is a probability distribution over action sequences. We have discussed its common choices in the paragraph starting from line 109 in the original manuscript. We added some clarifications on $g_\phi$ in the revision.
>
> > What is $d_a$ in line 93?
>
> `Re`: it’s the dimensionality of the action space introduced in line 74. We’ve added clarifications in the revision.
>
> > Theorem 3.1 is entirely broken …
>
> `Re`:
> Due to space constraints, we still couldn’t fit all details in the main paper, but we restructured the theorem in the revision which hopefully improves the readability. The main changes are:
>
> 1. We stated the key assumptions to our method and left the standard assumptions of CEM convergence to appendix.
> 2. We gave the sketch of the proof in the main paper, and left the full details to the appendix.
> 3. The appendix is updated accordingly.

---

### Author Response · Authors · 2022-08-02
**Summary of the revision**

Dear reviewers,

We have revised the main paper and the appendix, incorporating your feedback. Thanks for helping improve the quality of our work and here are the highlights of the revision:

1. We added experimental results:
    - We added a baseline DecentPETS (decentralized PETS). The learning curves in Fig. 6 and the discussion of it are updated accordingly.
    - We added a new section at the end of the appendix, which includes the visualization of planning trajectories of DecentCEM-A and POPLIN-A.
2. We restructured the theorem to improve the readability:
    - We stated the key assumptions of our method in the main paper, and left the standard assumptions of CEM convergence to the appendix.
    - We gave the sketch of the proof in the main paper, and left the full details to the appendix.
The appendix is updated accordingly.
3. We incorporated your writing suggestions.

---

### Meta-Review · Area_Chair_ETqv · 2022-08-27

**Recommendation:** Accept
**Confidence:** Certain

**Metareview:**

This paper proposes a parallelized version of the classic cross-entropy optimization method, by using an ensemble of CEM instances running independently from one another, and each performing a local improvement of its own sampling distribution. Both a theoretical and empirical analysis are provided to demonstrate the effectiveness of this simple decentralized approach. The reviewers find the paper to be overall well-presented, and appreciate the fact that the proposed is simple but effective. Consequently, this work can be utilized in any CEM-based model-based reinforcement learning algorithm.

**Award:**

No

---

### Decision · Program_Chairs · 2022-09-14

Accept